# Comparative Renal Safety of Tirzepatide and Semaglutide: An FDA Adverse Event Reporting System (FAERS)—Disproportionality Study

**DOI:** 10.3390/jcm14217678

**Published:** 2025-10-29

**Authors:** Ayush Gandhi, Nilay Bhatt, Alireza Parhizgar

**Affiliations:** 1Department of Hospital Medicine, The University of Texas MD Anderson Cancer Center, Houston, TX 77030, USA; 2Department of Internal Medicine, HCA Houston Healthcare, Houston, TX 77090, USA; docnilaybhatt@gmail.com; 3Department of Emergency Medicine, The University of Texas MD Anderson Cancer Center, Houston, TX 77030, USA; aparhizgar@mdanderson.org

**Keywords:** acute kidney injury, type 2 diabetes, GLP-1 receptor agonists, FDA Adverse Event Reporting System—disproportionality study, drug safety

## Abstract

**Background:** Acute kidney injury (AKI) remains a serious complication among individuals with type 2 diabetes. Glucagon-like peptide-1 receptor agonists (GLP-1 RAs) are widely prescribed and often regarded as kidney-protective, yet post-marketing reports have linked them to AKI. Tirzepatide, a newer dual GIP/GLP-1 agonist, shows well-documented metabolic benefits, but its renal safety in real-world use is not well characterized. **Methods:** We conducted a disproportionality analysis of the U.S. FDA Adverse Event Reporting System (FAERS) from January 2022 to September 2025. Reporting odds ratios (RORs) and proportional reporting ratios (PRRs) were used to compare AKI reporting between tirzepatide and semaglutide. **Results:** Among 133,872 reports (92,807 tirzepatide; 41,065 semaglutide), AKI was listed in 432 (0.47%) and 440 (1.07%) cases, respectively. The ROR for tirzepatide versus semaglutide was 0.44 (95% CI, 0.38–0.50), suggesting a lower reporting frequency for AKI with tirzepatide. **Conclusions:** In this real-world pharmacovigilance analysis, semaglutide but not tirzepatide showed a disproportionality signal for AKI. While causality cannot be confirmed, clinicians should ensure hydration and renal monitoring when initiating GLP-1 RAs, particularly semaglutide. Semaglutide showed a higher AKI reporting rate than tirzepatide, though these findings should be interpreted cautiously given reporting bias and potential confounders. Both agents appear safe, with low AKI frequency in practice. Further studies should determine if differences reflect biological or reporting effects. These findings support the need for larger epidemiologic studies to define risk modifiers and optimize clinical safety strategies.

## 1. Introduction

Patients with type 2 diabetes are at substantially higher risk of acute kidney injury (AKI) compared to the general population [1]. This is driven not only by diabetic nephropathy and vascular complications but also by the frequent presence of other comorbidities that predispose the kidney to injury. Obesity, which often overlaps with diabetes, introduces additional factors such as hypertension and fatty liver disease, further amplifying susceptibility. Episodes of AKI in this population carry serious consequences, including progression to end-stage kidney disease and greater cardiovascular risk [1]. For this reason, understanding how modern diabetes and weight-management therapies affect AKI risk is an important clinical question.

Glucagon-like peptide-1 receptor agonists (GLP-1 RAs) have become central to diabetes and obesity treatment because of their strong effects on glucose reduction, weight loss, and cardiovascular protection. Clinical trial data suggest that GLP-1 RAs improve chronic kidney disease (CKD) outcomes, for example, by reducing macroalbuminuria and slowing eGFR decline [2,3,4,5]. However, GLP-1 RAs are associated with a modestly increased risk of AKI compared to SGLT2 inhibitors in real-world data, but a lower risk compared to DPP-4 inhibitors, sulfonylureas, and basal insulin [6]. Most AKI events are attributed to prerenal mechanisms, such as dehydration from severe gastrointestinal side effects (nausea, vomiting, diarrhea), rather than direct nephrotoxicity [7,8,9]. Rare cases of intrinsic kidney injury, including interstitial nephritis, have been reported, but causality remains uncertain, and the absolute event rates in clinical trials are very low (<0.5%) [8,10].

Semaglutide is a widely used GLP-1 RA with proven benefits but also scattered reports of AKI. Case reports describe both prerenal azotemia and interstitial nephritis [11,12]. Pharmacovigilance studies confirm that semaglutide, along with liraglutide, has been associated with a modest signal for AKI in the FAERS database [13]. While clinical trial data show very low absolute event rates [14,15], this discrepancy between trial and real-world data warrants attention.

Tirzepatide, a dual GIP and GLP-1 agonist, was approved in 2022 and has shown impressive metabolic benefits. Post-hoc analyses of clinical trials suggest it may slow CKD progression [16,17], but real-world safety data are still limited. A few case reports describe AKI with tirzepatide use [18], and the drug’s label notes this as a potential risk. However, initial pharmacovigilance analyses have not identified a clear signal [18].

Because neither the SUSTAIN trials for semaglutide nor the SURPASS trials for tirzepatide were designed to capture rare renal events, head-to-head comparisons are lacking. To address this gap, we conducted a comparative FAERS disproportionality analysis of AKI reports for both drugs from January 2022 through September 2025. Although spontaneous reporting databases have limitations, they can shed light on whether one agent is disproportionately associated with AKI.

Globally, the prevalence of type 2 diabetes has exceeded 10% of the adult population, with renal complications remaining a leading cause of morbidity and mortality [19,20]. The COVID-19 pandemic further intensified metabolic and renal vulnerability in diabetic patients, with transient kidney injury frequently reported during acute infection [21]. These factors have renewed interest in incretin-based therapies, not only for glycemic control but also for their potential renal protection and improvement in quality of life under real-world conditions [22]. Studies have documented that patients treated with semaglutide, both oral and subcutaneous, often report improved treatment satisfaction and health-related quality of life compared to standard care [23].

In light of these considerations, we sought to systematically examine the renal safety profile of incretin-based therapies using real-world pharmacovigilance data. The primary objective was to compare the frequency of acute kidney injury (AKI) reports associated with tirzepatide and semaglutide in the FDA Adverse Event Reporting System. The secondary objectives were to describe the demographic characteristics and clinical outcomes of AKI reports for each drug.

## 2. Materials and Methods

The primary and secondary study objectives were defined as described in the Introduction. We conducted a retrospective pharmacovigilance study using the publicly accessible U.S. Food and Drug Administration (FDA) Adverse Event Reporting System (FAERS), covering the period from January 2022 through September 2025. All U.S. domestic reports were reviewed, and cases were included if tirzepatide or semaglutide was listed as the primary suspect drug. FAERS data were obtained directly from the U.S. Food and Drug Administration website and analyzed using Microsoft Excel (Version 16.78; Microsoft Corporation, Redmond, WA, USA). Acute kidney injury (AKI) was broadly defined using relevant MedDRA preferred terms, including acute kidney injury, renal failure, renal impairment, renal injury, tubulointerstitial nephritis, renal tubular necrosis, and anuria. These specific MedDRA Preferred Terms (PTs) were predefined and applied consistently to identify AKI-related reports. All PTs correspond to MedDRA version 26.0. Duplicate case entries were identified through the FAERS unique primary-ID and case-version system. When multiple versions of the same case were present, only the most recent version was retained to prevent overcounting. Reports lacking a valid case identifier were excluded. For each drug, we tabulated total adverse event (AE) reports, AKI counts, sex distribution, year of report, and serious outcomes (death, disability, hospitalization, life-threatening).

To assess whether AKI was disproportionately reported, we used two established disproportionality metrics. The reporting odds ratio (ROR) was calculated from 2 × 2 contingency tables as (a × d)/(b × c), where a represents AKI cases with the drug of interest, b non-AKI cases with the drug of interest, c AKI cases with the comparator drug, and d non-AKI cases with the comparator. Ninety-five percent confidence intervals (CIs) for the ROR were estimated using the standard error of the natural log of ROR. We also calculated the proportional reporting ratio (PRR), defined as [a/(a + b)]/[c/(c + d)]. For clarity, Table 1 summarizes the raw counts used to derive the disproportionality metrics. A disproportionality signal was considered present when the lower bound of the 95% CI for the ROR excluded 1, or when the PRR was ≥2 with at least three cases [24,25].

This study was conducted and reported in accordance with the STROBE (Strengthening the Reporting of Observational Studies in Epidemiology) statement [26], identified through the Equator Network [27]. A completed STROBE checklist is available in the Appendix A.

## 3. Results

We identified 92,807 domestic AE reports for tirzepatide, including 432 (0.47%) AKI cases, and 41,065 reports for semaglutide, including 440 (1.07%) AKI cases. Relative disproportionality favored tirzepatide: the ROR comparing tirzepatide vs. semaglutide was 0.44 (95% CI 0.38–0.50), with a PRR of 0.44. Accordingly, AKI appeared less frequently among tirzepatide reports than among semaglutide reports (Table 2).

Sex distribution among AKI cases differed between drugs. With tirzepatide (*n* = 432), 34.3% were male, 41.0% female, and 24.8% not specified; with semaglutide (*n* = 440), 38.9% were male, 56.4% female, and 4.8% not specified (Figure 1).

AKI reports rose steadily for both drugs, reaching 186 cases for tirzepatide and 239 cases for semaglutide in 2025 (Figure 2).

Serious outcomes among AKI reports were numerically higher for semaglutide: tirzepatide had 16 deaths, 6 disabilities, 209 hospitalizations, and 11 life-threatening events; semaglutide had 28 deaths, 11 disabilities, 243 hospitalizations, and 36 life-threatening events (Figure 3).

## 4. Discussion

This analysis demonstrates a marked divergence in pharmacovigilance signals between tirzepatide and semaglutide. Acute kidney injury (AKI) was reported at more than twice the proportional rate with semaglutide compared to tirzepatide, with a reporting odds ratio of 0.44 (95% CI, 0.38–0.50) favoring tirzepatide. The proportional reporting ratio confirmed this observation. Semaglutide, unlike tirzepatide, crossed conventional thresholds for a disproportionality signal.

Although the rise in AKI reports for both agents likely reflects broader clinical uptake, semaglutide consistently accounted for a larger share of serious outcomes, including hospitalizations and deaths. This pattern echoes earlier case reports and pharmacovigilance studies implicating GLP-1 receptor agonists in AKI, usually linked to gastrointestinal fluid loss, though biopsy-proven interstitial nephritis has occasionally been reported [11,13,28].

The lack of a disproportionality signal for tirzepatide should not be taken to mean no risk; rather, it suggests that reporting frequencies have not yet exceeded background expectations. In contrast, semaglutide’s enrichment for AKI aligns with prior pharmacovigilance data and supports its recognition as a GLP-1 RA with a reproducible, albeit infrequent, renal safety concern [2,13,18]. These findings suggest a potential safety distinction between tirzepatide and semaglutide in real-world reporting. Importantly, the absolute frequency of AKI with GLP-1 RAs remains very low in both trials and routine practice, generally below 1%. Thus, even though semaglutide shows a disproportionality signal, the clinical likelihood of AKI remains uncommon.

Clinical trial evidence provides an important counterpoint. In both the SURPASS program for tirzepatide and the SUSTAIN/PIONEER trials for semaglutide, AKI events were rare. A pooled analysis of semaglutide trials identified only 18 AKI cases among more than 9000 treated participants, mirroring comparator rates [15,29]. Similarly, cardiovascular outcome studies have suggested renal benefit, such as reduced macroalbuminuria with semaglutide [2,3,5] and slower eGFR decline with tirzepatide [16,17]. Small numerical imbalances in AKI events have been observed in semaglutide trials (8 vs. 1 with comparators), though these were not statistically significant [11,30]. Such findings may reconcile with the FAERS signal, suggesting that rare events emerge in practice despite being underpowered for detection in trials.

In clinical practice, these findings underscore the importance of patient counseling and monitoring [31]. For semaglutide, the reproducible AKI signal highlights the need to ensure adequate hydration, monitor renal function, and hold therapy temporarily during episodes of intercurrent illness [8]. Tirzepatide, despite not showing a signal to date, should be managed with similar precautions given the presence of isolated case reports [18]. Although uncommon, AKI can be clinically significant: a notable proportion of cases required hospitalization, and some resulted in death [13]. Encouragingly, renal function often recovered after drug withdrawal and rehydration, and rechallenge may be possible unless interstitial nephritis is suspected [32].

### 4.1. Mechanistic Considerations

Mechanistically, the most plausible cause of AKI with GLP-1 receptor agonists is prerenal azotemia from volume depletion driven by gastrointestinal fluid losses, particularly during early dose escalation [7,13,33]. These natriuretic and vasodilatory effects can transiently reduce effective renal perfusion in susceptible individuals, leading to hemodynamically mediated injury [34,35,36]. Rarely, idiosyncratic interstitial nephritis has been confirmed on biopsy [11,37,38]. In contrast, tirzepatide’s dual GIP component may attenuate these effects through improved tubular glucose handling and energy balance, although the evidence remains preliminary [2,39]. Further studies integrating mechanistic biomarkers with clinical outcomes will be essential to clarify these pathways.

### 4.2. Perspectives for Clinical Practice

While the observed divergence in AKI reporting between tirzepatide and semaglutide does not imply causality, it represents a hypothesis-generating signal with practical relevance. For patients initiating semaglutide, especially those with underlying kidney disease or on nephrotoxic co-medications, clinicians should adopt a proactive approach [7]. This includes gradual dose titration, close monitoring of renal function, and encouraging patients to maintain adequate hydration. Temporary treatment interruption during intercurrent illness or volume-depleting events can mitigate risk.

For tirzepatide, the current absence of a disproportionality signal is reassuring but not definitive. As its clinical use expands, rare renal events may still emerge, making continued pharmacovigilance and real-world monitoring essential [40].

Importantly, these findings highlight the value of integrating pharmacovigilance data with routine clinical decision-making. Real-world safety signals can help clinicians individualize therapy choices, balancing efficacy, patient comorbidities, and safety considerations. In practice, a structured monitoring plan, patient education on early signs of dehydration or renal stress, and coordination with multidisciplinary teams can help reduce preventable complications and optimize outcomes in diabetes care.

### 4.3. Reporting Dynamics and Market Exposure

Semaglutide’s broader use during the “Ozempic era,” including off-label prescriptions for weight loss, may have amplified its representation in FAERS reports. The influx of non-diabetic users, greater media visibility, and increased clinician awareness could partly explain the higher reporting frequency. In contrast, tirzepatide’s shorter market presence and narrower exposure window may contribute to fewer cumulative reports. These dynamics highlight how market exposure and social awareness, rather than intrinsic nephrotoxicity, can shape apparent pharmacovigilance trends.

A comparable trend was reported by Raičević BB et al. (2025) [41] in their analysis of serious adverse events with anti-obesity drugs, where market novelty and exposure duration were key determinants of reporting rates. Our findings parallel those observations, suggesting that similar factors may partly explain the observed differences between semaglutide and tirzepatide.

### 4.4. Study Limitations

This study has several important limitations that should be considered when interpreting the findings. First, the analysis is based on spontaneous reports submitted to the FAERS database, which are inherently subject to underreporting, selective reporting, and variable data quality. Duplicate submissions may occur despite internal safeguards, and reporting patterns can be influenced by factors such as media attention, regulatory actions, or evolving clinical familiarity with a drug.

Second, FAERS does not include reliable exposure denominators, so true incidence or risk estimates cannot be calculated. The findings reflect reporting proportions rather than population-level rates, which limits direct comparisons with clinical trial data.

Third, these results are derived from U.S. reports and may not be fully generalizable to other countries, where patterns of prescribing, patient characteristics, and pharmacovigilance practices may differ. They may also not reflect the experience of patients enrolled in randomized controlled trials, who typically represent more selected populations.

Finally, the role of confounding factors cannot be excluded. Coexisting illness, concurrent medications, or rapid dose escalation may contribute to renal events and are often incompletely captured in spontaneous reports.

These associations must be interpreted within the limits of spontaneous reporting data. The FAERS database does not account for comorbidities, concurrent medications, or dehydration from gastrointestinal side effects, which are major contributors to AKI in real-world settings.

Even with these limitations, FAERS analyses provide valuable hypothesis-generating insights that can complement prospective studies and help shape targeted research questions, post-marketing safety evaluations, and practical clinical vigilance [25,42].

## 5. Conclusions

This analysis highlights meaningful differences in the pattern of acute kidney injury (AKI) reporting between semaglutide and tirzepatide in real-world pharmacovigilance data. In this pharmacovigilance analysis, semaglutide but not tirzepatide showed a disproportionality signal for acute kidney injury (AKI). However, these findings should not be interpreted as proof of causal risk differences. The overall frequency of GLP-1 RA–RA-associated AKI in practice appears low and should be viewed as hypothesis-generating.

Clinicians should remain attentive to early signs of renal stress when initiating GLP-1 receptor agonists, particularly semaglutide. Patient education about hydration, temporary drug discontinuation during acute illness, and early laboratory monitoring can help reduce preventable renal complications. For tirzepatide, the absence of a signal so far is encouraging, but continued monitoring is needed as clinical exposure grows and more diverse populations begin treatment.

Beyond their immediate safety relevance, these findings illustrate how pharmacovigilance data can complement clinical trials and offer early insights into rare adverse events that might otherwise remain undetected. Future studies integrating FAERS data with electronic health records and prospective cohort analyses will be essential to clarify whether these reporting differences reflect a true biological distinction or variations in use and awareness.

In practical terms, both semaglutide and tirzepatide remain valuable agents with established metabolic and renal benefits. Their use should be guided by careful clinical judgment, informed monitoring, and an ongoing commitment to patient safety as new evidence continues to emerge.

## Figures and Tables

**Figure 1 jcm-14-07678-f001:**
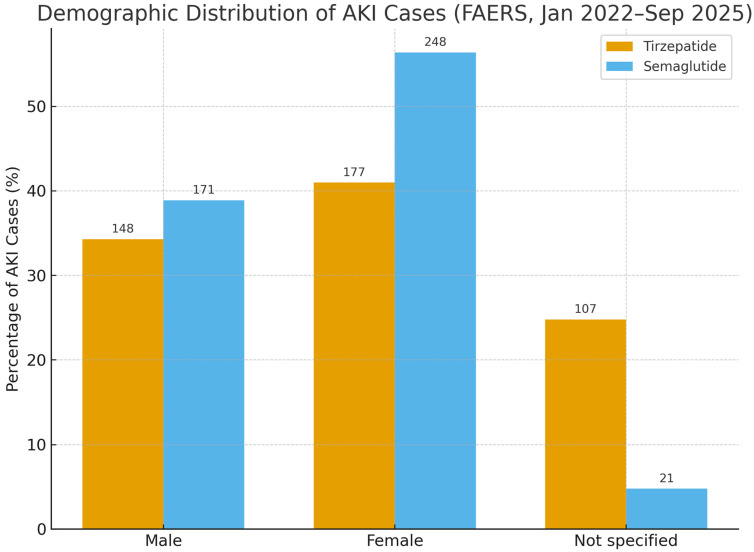
Demographic distribution of acute kidney injury (AKI) cases reported with tirzepatide (*n* = 432) and semaglutide (*n* = 440) in FAERS, January 2022–September 2025. Bars represent the percentage of AKI cases by sex. Absolute counts are displayed above each bar. No 95% CI is shown as this reflects descriptive reporting proportions. Abbreviations: AKI, acute kidney injury; FAERS, FDA Adverse Event Reporting System.

**Figure 2 jcm-14-07678-f002:**
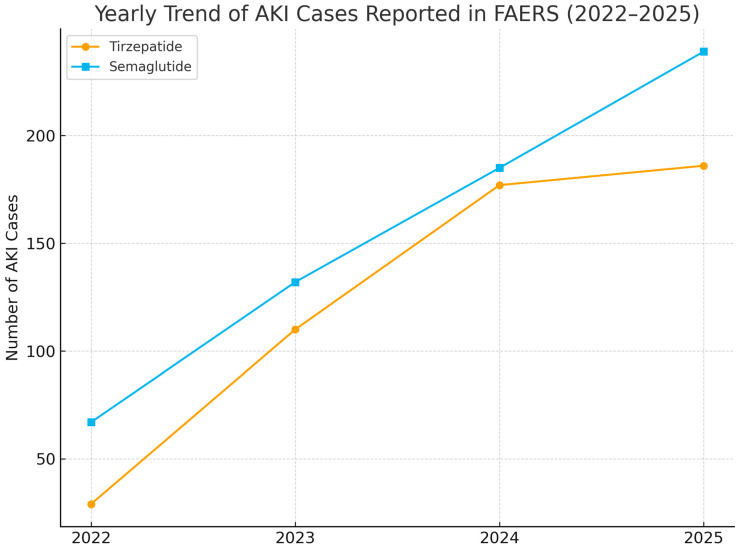
Yearly trend of acute kidney injury (AKI) cases reported with tirzepatide and semaglutide in the U.S. Food and Drug Administration Adverse Event Reporting System (FAERS) from 2022 through 2025. The *y*-axis shows total AKI case counts per year. No confidence intervals are presented as these represent raw reporting counts. Abbreviations: AKI, acute kidney injury; FAERS, FDA Adverse Event Reporting System.

**Figure 3 jcm-14-07678-f003:**
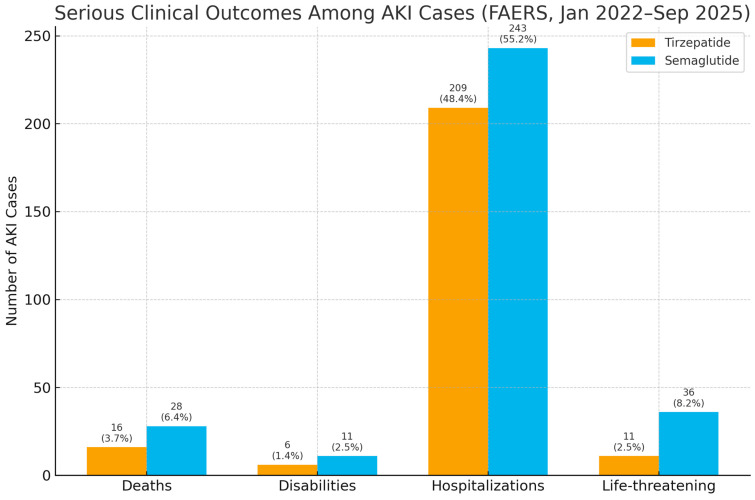
Serious clinical outcomes among acute kidney injury (AKI) cases reported with tirzepatide (*n* = 432) and semaglutide (*n* = 440) in the U.S. Food and Drug Administration Adverse Event Reporting System (FAERS, January 2022–September 2025). Bars represent the number and percentage of AKI cases. No confidence intervals are shown as these reflect descriptive proportions. Abbreviations: AKI, acute kidney injury; FAERS, FDA Adverse Event Reporting System.

**Table 1 jcm-14-07678-t001:** 2 × 2 contingency table showing counts for acute kidney injury (AKI) and non-AKI reports for tirzepatide (drug of interest) and semaglutide (comparator).

Drug	AKI (Event)	Non-AKI (No Event)	Total
Tirzepatide (Drug of Interest)	a = 432	b = 92,375	92,807
Semaglutide (Comparator)	c = 440	d = 40,625	41,065

Note: a, b, c, and d correspond to standard cell notation for disproportionality analysis, where ROR = (a × d)/(b × c) and PRR = [a/(a + b)] ÷ [c/(c + d)].

**Table 2 jcm-14-07678-t002:** Disproportionality analysis of acute kidney injury (AKI) reports for tirzepatide vs. semaglutide (FAERS, January 2022–September 2025).

Drug	Total AEs	AKI Cases	AKI Rate (%)	ROR (95% CI)	PRR
Tirzepatide	92,807	432	0.47	0.44 (0.38–0.50)	0.44
Semaglutide	41,065	440	1.07	Reference	Ref

Abbreviations: AE, adverse event; AKI, acute kidney injury; ROR, reporting odds ratio; PRR, proportional reporting ratio; CI, confidence interval. Note: ROR and PRR are computed directly from the counts shown above; no hypothesis test was performed.

## Data Availability

The data supporting this study are publicly available from the U.S. Food and Drug Administration Adverse Event Reporting System (FAERS) on the FDA’s official website.

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
