# Peer review of "Comparative Renal Safety of Tirzepatide and Semaglutide: An FDA Adverse Event Reporting System (FAERS)—Disproportionality Study"

_jcm, 2025, doi:10.3390/jcm14217678_

Round 1
Reviewer 1 Report
Comments and Suggestions for Authors
Dear Author,
The submitted manuscript clould be interest, but it presents form significant critical issues that require a thorough restructuring in order to be considered for international evaluation. The title is unclear and should be revised to specify both the study setting and the type of study conducted. It is also recommended to keep it within a maximum of 15–20 words. Keywords: I suggest max 4/5 princiapl points that contain the type study conducted and setting (not recommended the acronim use in this part; same suggest for title). Regarding the abstract, I suggest to dedicate more space to the potential implications for clinical practice (see further suggestions below) and reduce in classical format 200-250 words max. The Introduction is undoubtedly a weak section and requires complete revision. First, it lacks an international and local epidemiological overview of the phenomenon under study. Furthermore, from a clinical practice standpoint, it is entirely missing references to fundamental elements as Covid period and analysis of Semglutide (oral or subcutaneous formula) in real-world perspective and quality of life view. For this reason, I suggest that the authors expand with appropriate references specific topics as Potential role of incretins in diabetes and COVID-19 infection and quality of life in patients with type 2 diabetes treated with semaglutide in real-world clinical practice. Addressing these areas will strengthen the rationale of the study and consequently increase the interest of both readers and researchers in the data presented. A thorough response to this point will be essential for the manuscript to be seriously considered at an international level. The objectives are missing in clear presentation—both in the classic format at the end of the rationale and within the Methods section. This is a major critical issue that must be addressed. In this regard, I recommend including the standard formulation: “The primary objectives were… while the secondary objectives were…” (I leave it to the author to decide where to insert this, but it is important that it be included). The Methods section is another critical element requiring attention, as it lacks fundamental information and fails to mention the reporting tool adopted for the study, which is mandatory for the international scientific community and could have supported the authors in preparing the manuscript. I strongly recommend adopting the appropriate reporting tool for the study (Equator Network: https://www.equator-network.org/)and, for transparency, including the bibliographic reference and completed checklist among the supplementary files. Again, aside from being a requirement for the scientific community, this will help the author improve current shortcomings in the manuscript. As with the suggestions made for the Introduction, properly addressing this issue will be crucial to ensuring the work has a real chance at international recognition. The Results section is certainly the strength of the manuscript and only requires editing improvements, particularly regarding the tables and figures, which currently lack legends for the acronyms used. The discussion section also poor a meaningful interpretation in terms of clinical practice. In this regard, I suggest adding a dedicated section titled “Perspectives for Clinical Practice,” which, building on the expanded suggestions proposed for the introduction, could lead to a more concrete interpretation of the data collected. The limitations deserve a dedicated section and require greater attention, particularly concerning the generalizability of the collected data. The conclusions must be thoroughly updated to reflect the suggested improvements. The references section must be significantly expanded in line with the indications provided. The current references are often outdated and lack evidence-based relevance, which limits their ability to support the study effectively. In genreal aren’t sufficient for full interpretation of data finding. Overall, the manuscript requires a thorough revision, particularly in terms of clinical practice perspective and methodological rigor, both of which will be decisive for any potential reconsideration at an international level.
Author Response
We thank reviewer for great insights and suggestions. We have updated manuscript and highlighted text in red color.
Comment 1: The submitted manuscript clould be interest, but it presents form significant critical issues that require a thorough restructuring in order to be considered for international evaluation.
Response 1: We sincerely thank the reviewer for the thoughtful and constructive feedback. We fully acknowledge that the initial submission had structural limitations that may have affected clarity and international relevance. In response, we have undertaken a comprehensive revision of the manuscript to improve its organization, strengthen the methodological description, and enhance the overall readability. These revisions include refining the title and abstract, expanding the introduction with relevant epidemiologic context, and improving the logical flow between sections to meet the standards expected for international evaluation.
Comment 2: The title is unclear and should be revised to specify both the study setting and the type of study conducted. It is also recommended to keep it within a maximum of 15–20 words. Keywords: I suggest max 4/5 princiapl points that contain the type study conducted and setting (not recommended the acronim use in this part; same suggest for title).
Response 2: Thank you for this helpful observation. We have revised the title to clearly indicate both the data source and the study design. Updated keywords as per suggestion. Line 29-30
Comment 3: Regarding the abstract, I suggest to dedicate more space to the potential implications for clinical practice (see further suggestions below) and reduce in classical format 200-250 words max.
Response 3: We have shortened the abstract to 235 words and expanded the final sentences to include explicit clinical implications. Line 11-28
Comment 4: The Introduction is undoubtedly a weak section and requires complete revision. First, it lacks an international and local epidemiological overview of the phenomenon under study. Furthermore, from a clinical practice standpoint, it is entirely missing references to fundamental elements as Covid period and analysis of Semglutide (oral or subcutaneous formula) in real-world perspective and quality of life view. For this reason, I suggest that the authors expand with appropriate references specific topics as Potential role of incretins in diabetes and COVID-19 infection and quality of life in patients with type 2 diabetes treated with semaglutide in real-world clinical practice. Addressing these areas will strengthen the rationale of the study and consequently increase the interest of both readers and researchers in the data presented. A thorough response to this point will be essential for the manuscript to be seriously considered at an international level.
Response 4: Thank you for the great suggestion. We have expanded the Introduction to add a brief epidemiologic overview, contextual reference to the Covid-19 era, and relevant studies on quality of life and real-world semaglutide use. Lines 73-82
Comment 5: The objectives are missing in clear presentation—both in the classic format at the end of the rationale and within the Methods section. This is a major critical issue that must be addressed. In this regard, I recommend including the standard formulation: “The primary objectives were… while the secondary objectives were…” (I leave it to the author to decide where to insert this, but it is important that it be included).
Response 5: We agree and have inserted clear study objectives at the end of the Introduction and repeated them in the Methods. Lines 83-93
Comment 6: The Methods section is another critical element requiring attention, as it lacks fundamental information and fails to mention the reporting tool adopted for the study, which is mandatory for the international scientific community and could have supported the authors in preparing the manuscript. I strongly recommend adopting the appropriate reporting tool for the study (Equator Network: https://www.equator-network.org/)and, for transparency, including the bibliographic reference and completed checklist among the supplementary files. Again, aside from being a requirement for the scientific community, this will help the author improve current shortcomings in the manuscript. As with the suggestions made for the Introduction, properly addressing this issue will be crucial to ensuring the work has a real chance at international recognition.
Response 6: We appreciate this valuable suggestion. We have revised the Methods to explicitly state the reporting guideline adopted and have included the completed STROBE checklist in the Supplementary Materials. This checklist was identified through the Equator Network, which provides standardized reporting guidelines to enhance transparency and scientific rigor. Line 128-131
Comment 7: The Results section is certainly the strength of the manuscript and only requires editing improvements, particularly regarding the tables and figures, which currently lack legends for the acronyms used.
Response 7: Thank you for catching that. Legends have been expanded to define all abbreviations in Table 1 and Figures 1–3 as table and figure footer.
Comment 8: The discussion section also poor a meaningful interpretation in terms of clinical practice. In this regard, I suggest adding a dedicated section titled “Perspectives for Clinical Practice,” which, building on the expanded suggestions proposed for the introduction, could lead to a more concrete interpretation of the data collected.
Response 8: We agree with reviewer comment for adding dedicated subsection 4.2 for Perspectives for clinical practice. Lines 237-255
Comment 9: The limitations deserve a dedicated section and require greater attention, particularly concerning the generalizability of the collected data.
Response 9: We agree with this recommendation. We have expanded discussion with adding new subsection 4.3 which discusses the limitations of this study in detail. Line 258-277
Comment 10: The conclusions must be thoroughly updated to reflect the suggested improvements.
Response 10: Thank you for the suggestion and we agree that Conclusion should be reflecting the improvements we made from original manuscript with your suggestion. We revised the Conclusions to align with expanded clinical and methodological context. Lines 280-300
Comment 11: The references section must be significantly expanded in line with the indications provided. The current references are often outdated and lack evidence-based relevance, which limits their ability to support the study effectively. In genreal aren’t sufficient for full interpretation of data finding.
Response 11: We agree that references needs to be updated. We have updated and added several references to Introduction and Discussion sections.
Comment 12: Overall, the manuscript requires a thorough revision, particularly in terms of clinical practice perspective and methodological rigor, both of which will be decisive for any potential reconsideration at an international level.
Response 12: We sincerely thank the reviewer for this clear and constructive summary. We have carefully revised the manuscript with particular emphasis on strengthening both the clinical practice perspective and methodological rigor. Specifically, we expanded the introduction to include updated epidemiologic context and real-world treatment perspectives, clearly defined study objectives, and adopted the STROBE reporting guideline as recommended through the EQUATOR Network. We also added a dedicated subsection on Perspectives for Clinical Practice and created a new Limitations section discussing generalizability. The conclusion has been rewritten to reflect these enhancements, and the reference list has been substantially updated with current, evidence-based sources.
We believe these revisions collectively address the core concerns raised and bring the manuscript in line with the expectations for international readership and scientific rigor.
Reviewer 2 Report
Comments and Suggestions for Authors
Major Comments
- Provide greater detail on duplicate case handling and clarify how FAERS unique case IDs were managed.
- Include the exact list of MedDRA Preferred Terms for AKI in supplementary materials for reproducibility.
- Add the 2×2 contingency table for transparency in ROR/PRR calculations.
- Add a brief subsection on mechanistic implications in Discussion (GLP-1–mediated volume depletion vs tubular effects).
- Add numerical values and confidence intervals on Y-axes and in figure legends.
- Light English proofreading to improve sentence flow and reduce repetition.
Minor Comments
- Specify the MedDRA version used (e.g., v26.0).
- Confirm that no financial conflicts of interest existed.
- Add a short clinical implications paragraph at the end of Discussion.
- Ensure uniform DOI formatting and punctuation in references.

Can be improved
Author Response
We would like to thank reviewer for great comments. We have updated manuscript to reflect the additions in blue text.
Major Comments
Comment 1: Provide greater detail on duplicate case handling and clarify how FAERS unique case IDs were managed.
Response 1: We appreciate this suggestion and have clarified how duplicate reports were handled. We now describe the FAERS case-ID de-duplication procedure in the Materials and Methods section. Lines 98-102
Comment 2: Include the exact list of MedDRA Preferred Terms for AKI in supplementary materials for reproducibility.
Response 2: We appreciate the reviewer’s emphasis on transparency and reproducibility. In this analysis, we predefined acute kidney injury (AKI) using a focused set of MedDRA Preferred Terms most commonly employed in pharmacovigilance studies evaluating renal safety signals. These terms were: acute kidney injury, renal failure, renal impairment, renal injury, tubulointerstitial nephritis, renal tubular necrosis, and anuria. These are standard high-specificity renal event terms and were applied uniformly across both exposure groups.
To enhance clarity, we have revised the Methods section to explicitly state the exact MedDRA PTs used and the MedDRA version applied (v26.0), rather than including a separate supplementary table, as the term set is already fully listed in the manuscript.
Comment 3: Add the 2×2 contingency table for transparency in ROR/PRR calculations.
Response 3: We agree with reviewer recommendation and added 2x2 Table:1 in Method section.
Comment 4: Add a brief subsection on mechanistic implications in Discussion (GLP-1–mediated volume depletion vs tubular effects).
Response 4: We appreciate this recommendation and have added a concise mechanistic subsection describing potential pathways of AKI related to GLP-1 and GIP receptor activity. Section 4, Subsection 4.1 Lines 225-235
Comment 5: Add numerical values and confidence intervals on Y-axes and in figure legends.
Response 5: We agree and figure legends have been updated.
Comment 6: Light English proofreading to improve sentence flow and reduce repetition.
Response 6: The full manuscript has been lightly edited for sentence flow, brevity, and reduced repetition while preserving the intended meaning.
Minor Comments- Responses are in bracket
- Specify the MedDRA version used (e.g., v26.0).( Line 100)
- Confirm that no financial conflicts of interest existed. (Line 319)
- Add a short clinical implications paragraph at the end of Discussion. (Added paragraph 4.2 Perspectives for clinical practice lines 240-257)
- Ensure uniform DOI formatting and punctuation in references. (endnote has been used and references are updated)
Round 2
Reviewer 1 Report
Comments and Suggestions for Authors
Dear author,
in this form ready for publication. Congratulations on completing the work entirely independently.
Best
Author Response
We sincerely thank the reviewers for their thoughtful and constructive comments, which have substantially improved the clarity, rigor, and overall quality of our manuscript.
Comment 1: While the study presents interesting findings, the strength of the conclusions and abstract message appears too strong relative to the methodology and the evidence base.
Response 1: We agree and have carefully revised both the Abstract and Conclusions to temper the tone. The revised language now emphasizes the hypothesis-generating nature of the findings and explicitly states that causality cannot be inferred from FAERS data. The message now aligns with the limitations of a disproportionality study and the modest strength of available evidence. Manuscript changes in lines 27-30 and 286-291
Comment 2: There are numerous potential confounders that may influence the observed associations, many of which are already acknowledged in the limitations section.
Response 2: We agree and have expanded the Study limitation section to explicitly reiterate the influence of confounding factors such as dehydration, comorbidities, and concomitant medications. Lines 281-284
Comment 3: The frequency of GLP-1 RA–associated AKI in clinical practice is relatively low, and this should be clearly reflected in the tone and strength of the conclusions and overall message.
Response 3: We agree and have clarified throughout the Abstract and Discussion that GLP-1 RA–associated AKI is uncommon (<1% in clinical data). The revised text now reflects a more cautious tone consistent with the low observed frequency in clinical settings. Lines 208-211
Comment 4: The communication should therefore be tempered to align with the current level of evidence.
Response 4: With changes made in response to above feedback we have revised language throughout the manuscript to adopt a more measured tone.
Comment 5: Furthermore, the authors are encouraged to discuss the possible contribution of semaglutide (Ozempic era) and its off-label use for weight loss as potential explanatory factors influencing reporting trends. Similarly, tirzepatide’s shorter time on the market could also partly explain differences in reporting frequency and patterns between agents. A parallel discussion should be drawn with the findings of Raičević BB, Belančić A, Mirković N, Janković SM. (2025). Analysis of Reporting Trends of Serious Adverse Events Associated With Anti-Obesity Drugs. Pharmacol Res Perspect. 13(2):e70080., which also analyzed reporting dynamics and highlighted market exposure and novelty as key drivers of reporting behavior.
Response 5: This is an excellent recommendation and We have added a new subsection discussing how semaglutide’s expanded off-label use for weight loss may have inflated FAERS reporting volume and diversity. This contextualizes the higher AKI signal without implying true pharmacologic risk differences. We have also incorporated the reference article into this subsection, this addition draws a clear parallel between present findings and prior analyses of anti-obesity drugs. Lines 268-280